# Growth Behaviors of GaN on Stripes of Patterned c-Plane GaN Substrate

**DOI:** 10.3390/nano12030478

**Published:** 2022-01-29

**Authors:** Peng Wu, Jianping Liu, Lingrong Jiang, Lei Hu, Xiaoyu Ren, Aiqin Tian, Wei Zhou, Masao Ikeda, Hui Yang

**Affiliations:** 1Suzhou Institute of Nano-Tech and Nano-Bionics, Chinese Academy of Sciences, Suzhou 215123, China; pwu2018@sinano.ac.cn (P.W.); lrjiang2016@sinano.ac.cn (L.J.); lhu2017@sinano.ac.cn (L.H.); xyren2014@sinano.ac.cn (X.R.); aqtian2012@sinano.ac.cn (A.T.); wzhou2015@sinano.ac.cn (W.Z.); mikeda2013@sinano.ac.cn (M.I.); 2School of Physical Science and Technology, ShanghaiTech University, Shanghai 201210, China; 3Shanghai Advanced Research Institute, Chinese Academy of Sciences, Shanghai 201210, China; 4University of Chinese Academy of Sciences, Beijing 100049, China; 5Key Laboratory of Nanodevices and Applications, Chinese Academy of Sciences, Suzhou 215123, China

**Keywords:** step motions, spiral growth, island growth, patterned substrate

## Abstract

Growth behaviors of GaN on patterned GaN substrate were studied herein. Spiral and nucleation growth were observed after miscut-induced atomic steps disappeared. The morphology of nucleation growth at different temperature is explained by a multi-nucleation regime introducing critical supersaturation. Simulated results based on a step motion model successfully explain the growth behaviors on stripes. These findings can be applied to control the surface kinetics of devices such as laser diodes grown on patterned substrate.

## 1. Introduction

Due to their outstanding characteristics such as tunable direct wide band-gap and high electron saturation velocity, III-nitrides are widely used in optoelectronic and electronic devices. An atomically smooth surface is crucial for obtaining high internal-quantum-efficiency quantum wells [1] and high electron mobility transistors that enable efficient carrier transport [2]. In the growth of III-nitrides, smooth surfaces with uniformly-arranged atomic steps can be obtained by using substrates with an optimized miscut and can be maintained by realizing step-flow growth [3,4,5]. However, some step-instability-related rough surfaces with morphologies such as ridges [6] and fingers [7] often occur and deteriorate the surface. Hence, further studies on the kinetics of atomic steps are necessary to obtain a sharp interface.

Patterned substrates were firstly used to reduce the threading dislocation (TD) density of GaAs grown on silicon by the well-known epitaxial lateral overgrowth (ELOG) technology [8]. This technology did not attract much attention until it was successfully applied to InGaN-based blue laser diodes grown on ELOG sapphire substrates [9]. Researchers focused mainly on the quality [10,11] and adatom incorporation [12,13,14] of laterally grown material. A novel design of InGaN-based laser diodes grown on stripes of Si substrates shows great prospects [15]. A stripe on a patterned substrate is a good platform to study step motions. It can provide a finite surface and free-of-source atomic steps at the upstream edge. The growth on the stepped surface of stripes proceeds as follows [16,17,18]: (i) miscut-induced atomic steps move forward and the upper terrace width of the topmost step grows wider and wider; (ii) new step sources occur on the wide terrace due to screw dislocations or nucleation. In this work, we studied the growth behaviors of GaN grown on stripes of patterned GaN substrates. The morphologies of samples grown under different temperature were compared. The nucleation growth regime was explained and simulated by a step motion model.

## 2. Materials and Methods

GaN substrates with TD density of 10^6^–10^7^ cm^−2^ were chemo-mechanically polished into vicinal wurtzite (0001) c-plane with a miscut angle of about 0.35° towards $\left[1\bar{1}00\right]$ m-direction. After photolithography and ICP etching, the substrates were patterned into periodically arrayed rectangles which uniformly covered the whole GaN substrate. Each rectangle includes stripes along the $\left[11\bar{2}0\right]$ a-direction, stripes along the m-direction and unpatterned area. The detailed structure of the rectangle pattern is showed in Figure 1.

Afterwards, the samples were dipped into 25% tetramethylammonium hydroxide (TMAH) at 85 °C for 20 min to remove the etching residuals on the surface and make a smoother sidewall [19]. About 100 nm un-intentionally doped GaN was then grown by metalorganic chemical vapor deposition (MOCVD) at temperature of 993 °C (sample A) and 888 °C (sample B). Trimethylgallium (TMGa) and ammonia were used as the gas sources, while nitrogen was used as the carrier gas.

The surface morphology was studied by atomic force microscope (AFM, Bruker Dimension ICON) in tapping-mode. Large-scale AFM amplitude images were used to clearly show atomic step features on the surface with huge difference in height [20]. The miscut angles of GaN substrates were obtained by the analysis of the peak positions of X-ray rocking curves (XRD, Bruker D8 Discover) by rotating the samples around the surface normal.

## 3. Results

Both sample A and B show step-flow morphology with uniformly arranged straight atomic steps on the unpatterned area as shown in Figure 2. The miscut angles can be calculated to be 0.33° for sample A and 0.4° for sample B according to the average terrace width, which agrees well with the XRD measurements. After 100 nm GaN regrowth, miscut-induced atomic steps move laterally (along the m-direction) with a distance of about 17.4 μm for sample A and 14.2 μm for sample B. Hence, none of the miscut-induced steps are left on the 10 μm wide stripes along the a-direction. Screw dislocations are inevitable on such a large surface with an area of 2100 μm^2^ and become new step sources, resulting in a surface full of curved steps as shown in Figure 3.

For stripes along the m-direction, miscut-induced steps move freely except for the area of the upstream $\left[\bar{1}100\right]$ edge. On the upstream edge of stripes along m-direction, different morphologies with (as shown in Figure 4) and without (as shown in Figure 5 and Figure 6) the effect of screw dislocations are observed. The left sides of the dashed blue lines in Figure 4a,b, both show spiral growths which are originated from screw dislocations. This kind of spirals are caused by a pair of screw TDs of similar signs which are closely located [21]. The average interstep distance of curved steps is 155 nm for sample A and 126 nm for sample B. Uniform step-flow growth as shown in Figure 2 occurs on the right side of the dashed blue lines for both samples.

Figure 5 shows nucleation morphology consisted of tens of layers of rectangle islands on sample A. In the direction of $\left[1\bar{1}00\right]$ m and $\left[\bar{1}100\right]$ ₋m, it shows almost-bunched bilayer island edges. This kind of phenomenon is caused by different growth rates of step edges with one (type A) or two (type B) dangling bonds [22]. Step edges of type A and B alternately occur in the equivalent direction of m due to the atomic configuration of wurtzite c-plane nitrides [23]. Taking a closer look, the edges of the topmost island (with dimensions of 8.4 × 3.3 μm^2^) also show asymmetric kinetic behaviors. Its step edge in the m-direction moved very fast and almost caught up with the step of the lower layer. However, the step in the ₋m-direction was far behind. Hence, the step edge in the m-direction of the topmost island should be type B with two dangling bonds while that in the ₋m-direction is type A with one dangling bond.

Except for some pits, a large-area (17 × 7.3 μm^2^) smooth surface is obtained on the left side of the 10 μm-wide stripe along the m-direction in Figure 6a for sample B. According to Figure 6b, the larger pits are formed by edge TDs, while the smaller ones may be caused by the desorption of bulk atoms due to thermal instabilities [24]. Figure 6c shows the amplification at the intersection between the step-free surface and miscut-induced steps. As it shows, terrace width gradually narrowed (from 258 nm to 47 nm) to be equal to the width of steps induced by miscut. The atomic steps in Figure 6c were verified to be monolayer (~0.25 nm) high according to the height profile (not shown here).

## 4. Discussion

Many researchers have reported that there is competition between atomic steps induced by miscut and screw dislocations [3,25]. Under the conditions of a large miscut angle, step-flow growth prevails along with the existence of pined steps around small pits at the apex of the screw dislocations. On the contrary, under the conditions of a small miscut angle, large-area curved steps or even hillocks occur [26]. In our experiments, the miscut angles for both samples are large enough to suppress the growth around screw dislocations, as the morphology on the unpatterned area shows. However, for the patterned stripes, when steps induced by miscut disappear on the upstream edge of the stripes after a period of growth, spiral growth occur as shown in Figure 3 and Figure 4. Based on the Burton–Cabrera–Frank theory, the supersaturation (σ) can be derived from the interstep distance of growth spirals [27,28,29]. Thus, σ increases from 0.119 for sample A to 0.160 for sample B with reduced temperature, which is consistent with the results of Liu et al. [30]. Beyond that, there is no significant difference in the morphology of the growth spirals at different growth temperatures.

On the other hand, without the interference of screw TDs, the morphology is entirely different for these two samples. To investigate the morphological evolution on the screw-dislocation-free surface of stripes along the m-direction, we propose a quantitative step-by-step kinetic model based on the step motion model proposed by Schwoebel and Shipsey [31]. As shown in Figure 7, step velocity is simplified and expressed as
(1)$$v_{k}=C\left(s_{k}k_{+}+s_{k-1}k_{-}\right)$$

Here, C is a constant of the step motion which is equal for all steps at specific growth conditions, while s_k_ stands for the terrace width between the kth and (k+1)th steps. k_+_ and k_−_ are the probability of adatoms incorporating to the upper and lower steps, respectively. The Ehrlich–Schwoebel barrier (ESB), which describes the difference of energy barrier between the upper and lower terrace, is taken into account. For a single step, if the energy barrier for incorporation of the upper terrace is larger than that of lower terrace (k_+_ > k_−_), a positive ESB exists; otherwise, it is a negative ESB. To make this model more applicable, the following issues are considered:(a)The probability of adatom incorporation for steps with type A and B edges is different;(b)When the distance of adjacent steps is shorter than a critical value, step bunching occurs and bunched steps will move with the same speed;(c)If the terrace width is larger than the atomic diffusion length, only adatoms near steps (the distance from the step being shorter than the diffusion length) can feed the steps.

Based on the above considerations, we simulate the step kinetics on the upstream edge of the stripe along the m-direction without nucleation. As is shown in Figure 8a, steps are easily bunched to form macro steps several atomic monolayers high with a negative ESB. In the case of a positive ESB shown in Figure 8b, stable bilayers (typical step-flow morphology as shown in Figure 2) are formed due to atomic configuration asymmetry. Hence, the type of ESB in our experiments should be positive. Under a positive ESB, step velocity mainly depends on the width of the lower terrace. Steps move slower and slower with a gradually narrower width of the lower terrace moving towards the stripe edges, which increases the difficulty in accomplishing a full atomic layer. Hence, a morphology of narrower and narrower terraces near stripe edges occurs as shown in Figure 5 and Figure 6c.

According to Equation (1), the velocity of the upmost miscut-induced step will not be slower than that of the lower steps. Therefore, steps shown in Figure 6c, with a terrace width wider than that of the miscut at the border of the step-free and step-flow area, are formed by newly formed islands. In the classical homogeneous nucleation theory, there are two types of nucleation growth: single nucleation (a single island occurs and laterally spreads to cover the whole surface) and multi-nucleation (many islands nucleate at the same time and then coalesce to form a full layer). Single nucleation only happens on a very small facet while multi-nucleation prevails in most cases, and this always holds true [32]. The process of coalescence is so fast that it almost simultaneously takes place with the beginning of nucleation.

However, such ideal layer-by-layer growth did not occur. Multilayers of islands with gradually reduced terrace width near the edge came up instead. To explain this phenomenon, we introduce a critical supersaturation value σ_cri_ when the energetic barrier of nucleation is overcome [33]. On the step-free surface of stripes, σ is high in the center and low at the edges due to the gas source consumption for lateral growth of the sidewalls or vertical growth of miscut-induced step-flow in the m-direction. Supersaturation increases with an enlarged area of step-free surface. When σ in the middle region reaches a critical value (σ_cri_), multi-nucleation occurs and quickly coalesces to form a two-dimensional (2D) island of the size d_2D_. The growth condition at a high growth temperature promotes the lateral growth of the (11–20) sidewall [34] and therefore aggravates the non-uniform distribution of supersaturation on the stripe surface as shown in Figure 9. Nucleation occurs more readily on stripes at a higher growth temperature (smaller R_cri_ for sample A). When two samples both meet the conditions for nucleation, the size of the 2D island is larger for the sample with more uniform distributions of supersaturation (larger d_2D_ for sample B).

In our model, we assume that the condition for nucleation is met when the size of the step-free area reaches a critical value. Therefore, the nucleation processes on stripes is described as: (a) miscut-induced atomic steps flow downwards and a step-free surface (of the size L_SF_) forms at the upstream of the stripes; (b) multi-nucleation occurs and coalesces to form a 2D island (of the size d_2D_) at the middle of the surface when L_SF_ reaches a critical nucleation value (R_cri_); (c) the 2D island laterally spreads based on the step motion theory; (d) another new layer starts to grow when the size of the first island reaches the critical nucleation value R_cri_. Considering multi-nucleation processes, the simulated results are shown in Figure 10 with proper parameters. The simulated surface profiles of Figure 10a and 10b agree well with the morphology of samples A and B, respectively.

## 5. Conclusions

We have investigated the surface kinetics on stripes of patterned GaN substrates. Both spiral and nucleation growth occur when miscut-induced steps flow downwards. Multi-islands morphology is caused by the edge effect and a positive ESB on the stripes. Without the interference of screw TDs, the different morphologies at different growth temperatures are explained by a multi-nucleation growth regime under non-uniform distributions of surface supersaturation. Simulated results based on the step motion model further verify the growth process on stripes. These findings show the importance of controlling the surface kinetics in GaN growth and can be applied to control surface kinetics in devices such as laser diodes grown on patterned substrates.

## Figures and Tables

**Figure 1 nanomaterials-12-00478-f001:**
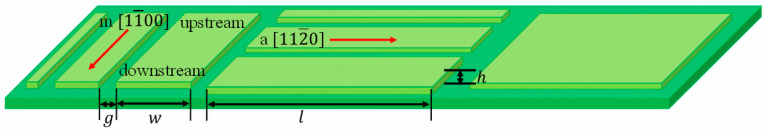
Schematic perspective view of the patterned GaN substrates, in which g (=20 μm) is the spacing of stripes, w (=2/5/10/15/30/50 μm) is the stripe width, l (=210 μm) is the stripe length and h (=0.5 μm) is the stripe height, respectively. The patterns include three regions: stripes along the m-direction (**left**), stripes along the a-direction (**middle**) and an unpatterned area (**right**).

**Figure 2 nanomaterials-12-00478-f002:**
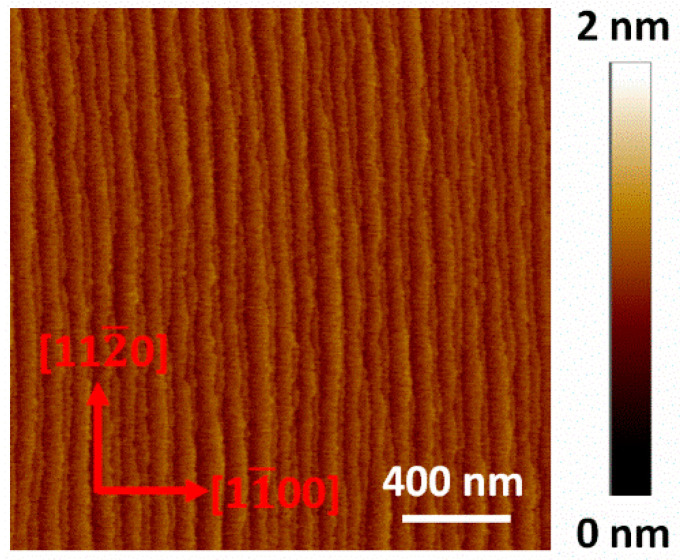
AFM height images (2 × 2 μm^2^) on the unpatterned area of sample A.

**Figure 3 nanomaterials-12-00478-f003:**
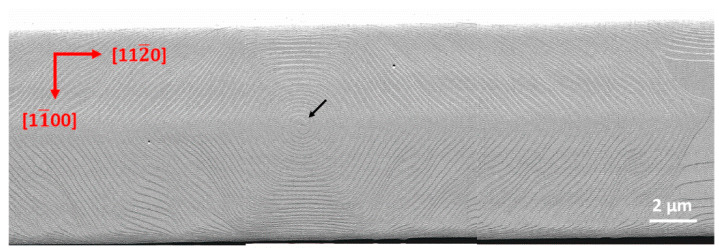
AFM amplitude images (30 × 10 μm^2^) on the stripes along the a-direction of sample A. The black arrow points to the apex of a screw dislocation.

**Figure 4 nanomaterials-12-00478-f004:**
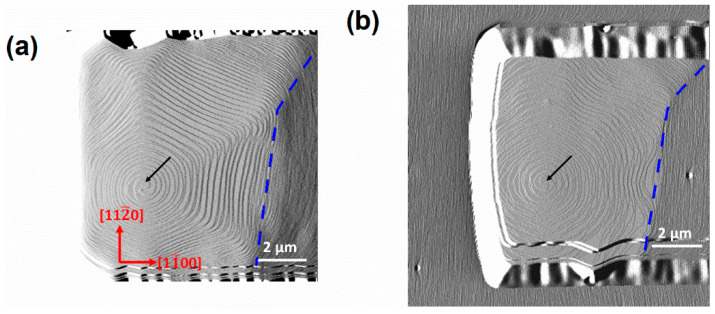
AFM amplitude images on the upstream edge of the 10 μm-wide stripe along the m-direction for sample A: (**a**) 10 × 10 μm^2^ and for sample B: (**b**) 12 × 12 μm^2^. The black arrows point to the apex of screw dislocations.

**Figure 5 nanomaterials-12-00478-f005:**
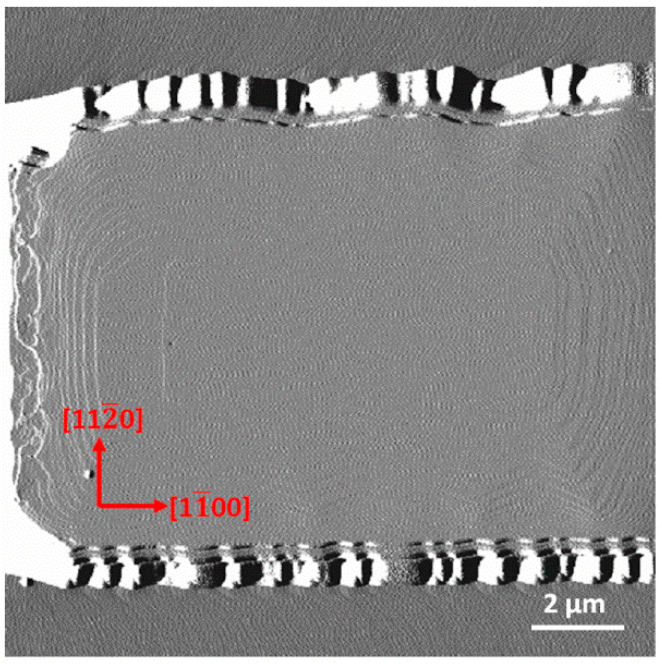
AFM amplitude images (14 × 14 μm^2^) on the upstream edge of 10 μm-wide stripe along the m-direction for sample A.

**Figure 6 nanomaterials-12-00478-f006:**
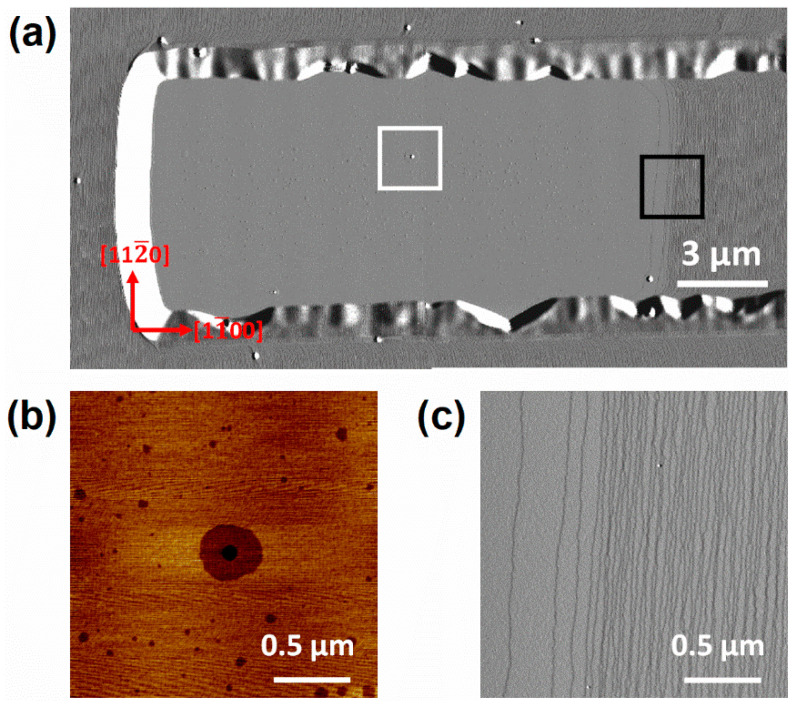
Large-scale AFM amplitude images (24 × 12 μm^2^) on the upstream edge of 10 μm-wide stripe along the m-direction for sample B (**a**) and its 2 × 2 μm^2^ amplification image; (**b**) height image of white square; (**c**) amplitude image of the black square.

**Figure 7 nanomaterials-12-00478-f007:**
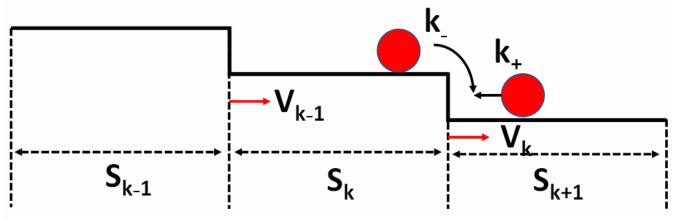
Step motion model on a vicinal surface. v_k_ is the velocity of the kth step while the widths of its upper and lower terrace are s_k_ and s_k−1_, respectively. k_+_ and k_−_ represent adatom incorporation probability from lower and upper terraces, respectively. v_k+1_ and s_k+1_ represent the velocity of the (k+1)th step and the width of its lower terrace.

**Figure 8 nanomaterials-12-00478-f008:**
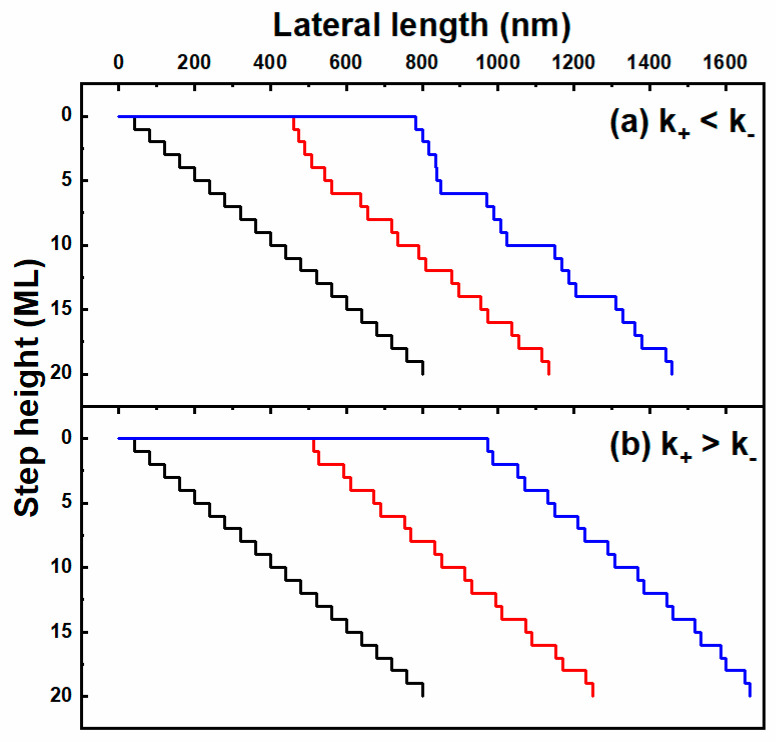
Surface profiles of morphological evolution on the upstream edge of stripes along the m-direction ignoring nucleation in the presence of negative ESB (**a**) and positive ESB (**b**). The different color lines show different growth stages (transition from black to red and finally blue).

**Figure 9 nanomaterials-12-00478-f009:**
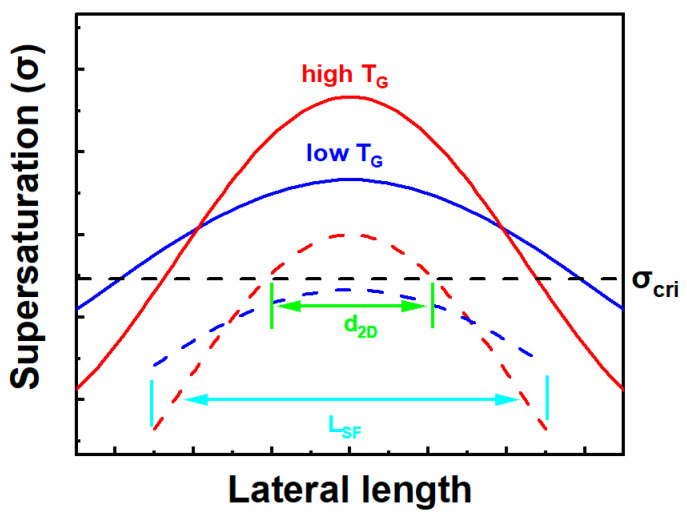
Surface supersaturation distribution on the step-free surface at different temperatures (red lines denote high growth temperature and blue lines denote low growth temperature) and different dimensions (solid lines represent large area and dashed lines represent small area). The x axis (lateral length) is along the m-direction at the center of the step-free surface.

**Figure 10 nanomaterials-12-00478-f010:**
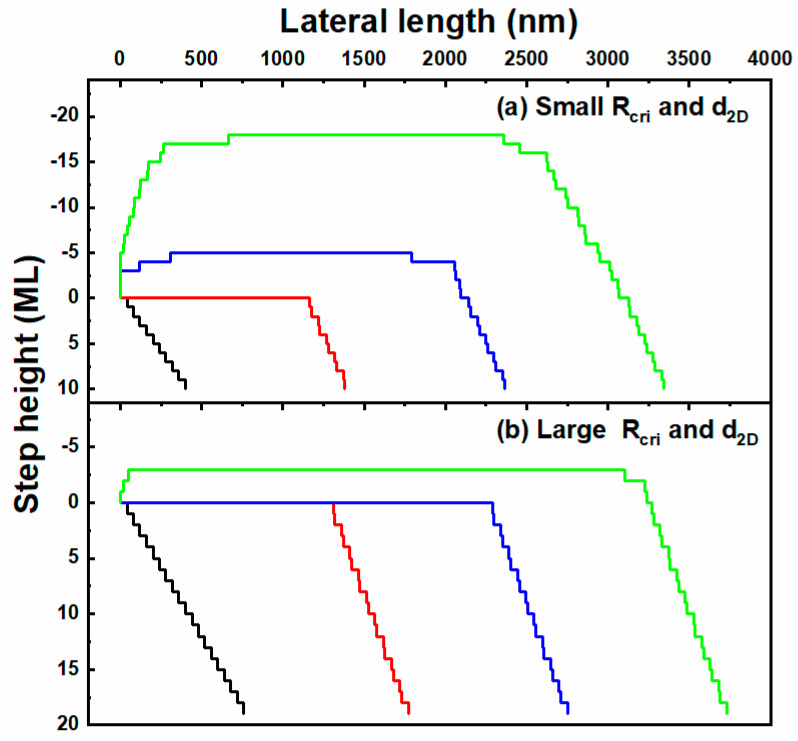
Simulated surface profiles of the nucleation growth on stripes at conditions of small (**a**) and large (**b**) R_cri_ and d_2D_.

## Data Availability

The data presented in this study are available on request from the corresponding author.

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
