# Peer review of "Growth Behaviors of GaN on Stripes of Patterned c-Plane GaN Substrate"

_nanomaterials, 2022, doi:10.3390/nano12030478_

Round 1

Reviewer 1 Report

I do not want to judge if these findings are interesting or not nor if they are important or not. However, what I want to say is, that this manuscript would strongly benefit when it would be corrected by a native speaker or a professional editing service. In addition, I think that both, the abstract and especially the conclusions should be more extensive. There are a few things which should be definitely changed:

  • In the Introduction: what is the meaning of high-quantum -efficiency quantum wells?
  • Figure 1 – font size is too small.
  • Figure 3 – hard to see anything from this figure. Could the contrast be increased?
  • Figure 4 and 5 should be presented larger, such that the minimum font size in the figure fits to the font size in the main text. Especially in Figure 5 almost nothing can be seen.
  • Figure 6 a and b have to be shown in a way that the details can be seen.
  • Discussion – the abbreviation BCF has to be introduced.
  • The abbreviation E-S barriers has to be introduced.
  • In the title and in the abstract, it is highlighted that “free standing” substrates are used. However, in the whole main text it is neither explained what is meant by that nor “free-standing” is used. Thus, why is this written in the title?

Author Response

Response to Reviewer 1 Comments

Point 1: This manuscript would strongly benefit when it would be corrected by a native speaker or a professional editing service. In addition, I think that both, the abstract and especially the conclusions should be more extensive.

Response 1: Thank you for your kind advice on English language and style. The revised manuscript has been corrected according to the advices of experts from MDPI English editing service. Meanwile, we have further improved the contents of abstract and conclusion. Kindly refer to the revised manuscript for details.

Point 2: There are a few things about figures and abbreviation which should be definitely changed…

Response 2: It is so kind of you to point out these mistakes in our manuscript. Firstly, to be more precise, “high-quantum-efficiency quantum wells” has been corrected to “high internal-quantum-efficiency quantum wells”. Secondly, we have adjusted the size and font size of Figure 1, 3, 4, 5, and 6. Besides, the contrast of Figure 3 has been increased. Please kindly examine if details in all figures can be seen. Thirdly, the full name of abbreviations of BCF and E-S barrier have been supplemented in the revised version. We also add a brief explanation of E-S barrier in the revised manuscript. Fnally, “freestanding substrate” means homoepitaxy on bulk substrate. To reduce misunderstanding, we directly use “GaN substrate” in place of “freestanding substrate” in the revised manuscript.

We hope the revised manuscript is now acceptable to you. If not, we are glad to receive any further feedback which we shall continue to apply our best effort to adress.

Reviewer 2 Report

This report deals with the growth of GaN and surface dynamics. Starting from quite simple structures (mesa aligned along 2 directions), authors can deduced interesting surface physics, supported by modelling. The nano features are related to details of the steps and atom incorporation: hence this paper deserves publication in nanomaterials.

A few errors should be corrected:

the following sentence is not very clear and should be clarified: Besides it is worth noting that the distance between islands tends to be shorter and shorter near the edge of stripes and the border between islands and straight steps induced by miscut.

The process of coalescence is so fast that (it) almost simultaneously take(s) place with the beginning...  

Dis-location

atomics steps

Figure 9: what is the x axis ? isn't it the position on the step free surface, with xmin and xmax being the edges of the free step surface ? I cannot understand the figure otherwise 

Author Response

Response to Reviewer 2 Comments

Point 1: The following sentence is not very clear and should be clarified: Besides it is worth noting that the distance between islands tends to be shorter and shorter near the edge of stripes and the border between islands and straight steps induced by miscut.

Response 1: This sentence is puzzling indeed. Hence, we delete it and add sentences with similar meaning at the end of paragraph above Figure 8. The supplemented sentences are “Under a positive ESB, step velocity mainly depends on the width of lower terrace. Steps move slower and slower with a gradually narrower width of lower terrace moving towards stripe edges, which increases the difficulty in accomplishing a full atomic layer. Hence, a morphology of narrower and narrower terraces near stripe edges occurs as shown in Figure 5 and 6c.”

Point 2: A few errors should be corrected…

Response 2: It is so kind of you to point out these mistakes in our manuscript. These grammar and spelling mistakes have been corrected using “Track Changes”. Please kindly refer to the revised manuscript in detail.

Point 3: Figure 9: What is the x axis ? isn't it the position on the step free surface, with xmin and xmax being the edges of the free step surface ? I cannot understand the figure otherwise.

Response 3: The x axis in Figure 9 is definitely the line along [1-100] direction across the center of step-free surface. We have modified the figure and caption to make it easily understood.

Please kindly check if the revised manuscript meet with approval. Once again, thank you very much for your comments and suggestions.

Round 2

Reviewer 1 Report

The authors have improved the manuscript and thus it can be published as it is.